# Zebrafish HERC7c Acts as an Inhibitor of Fish IFN Response

**DOI:** 10.3390/ijms24054592

**Published:** 2023-02-27

**Authors:** Yi-Lin Li, Xiu-Ying Gong, Zi-Ling Qu, Xiang Zhao, Cheng Dan, Hao-Yu Sun, Li-Li An, Jian-Fang Gui, Yi-Bing Zhang

**Affiliations:** 1State Key Laboratory of Freshwater Ecology and Biotechnology, Institute of Hydrobiology, Chinese Academy of Sciences, Wuhan 430072, China; 2University of Chinese Academy of Sciences, Beijing 100049, China; 3The Innovation Academy of Seed Design, Chinese Academy of Sciences, Wuhan 430072, China

**Keywords:** small HERC family, HERC7 subfamily, negative regulation, IFN response, protein attenuation

## Abstract

In humans, four small HERCs (HERC3-6) exhibit differential degrees of antiviral activity toward HIV-1. Recently we revealed a novel member HERC7 of small HERCs exclusively in non-mammalian vertebrates and varied copies of *herc7* genes in distinct fish species, raising a question of what is the exact role for a certain fish *herc7* gene. Here, a total of four *herc7* genes (named HERC7a–d sequentially) are identified in the zebrafish genome. They are transcriptionally induced by a viral infection, and detailed promoter analyses indicate that zebrafish *herc7c* is a typical interferon (IFN)-stimulated gene. Overexpression of zebrafish HERC7c promotes SVCV (spring viremia of carp virus) replication in fish cells and concomitantly downregulates cellular IFN response. Mechanistically, zebrafish HERC7c targets STING, MAVS, and IRF7 for protein degradation, thus impairing cellular IFN response. Whereas the recently-identified crucian carp HERC7 has an E3 ligase activity for the conjugation of both ubiquitin and ISG15, zebrafish HERC7c only displays the potential to transfer ubiquitin. Considering the necessity for timely regulation of IFN expression during viral infection, these results together suggest that zebrafish HERC7c is a negative regulator of fish IFN antiviral response.

## 1. Introduction

The HERC family proteins are E3 ubiquitin ligases that contain one C-terminal HECT (homologous to E6AP carboxyl terminus) domain and one or more N-terminal RCC1 (regulator of chromosome condensation 1)-like domains (RLD) [1]. In humans, the HERC family has six members that are traditionally classified into two subgroups: large HERCs (HERC1 and HERC2) and small HERCs (HERC3–6). Compared with large HERC proteins bearing one HECT domain, more than one RLD domain and multiple other conserved regions, small HERC proteins harbor a single HECT and a single RLD domain [2]. Such a structural difference supports the notion that large HERCs and small HERCs arise from convergent evolution of ancestors belonging to two distant families [3].

The E3 ubiquitin ligase activities of HERC proteins are ascribed to their C-terminal HECT domains that are capable of transferring ubiquitin to target proteins, a process of protein translational modification (PTM) termed ubiquitinylation, which has been verified in HERC1, HERC3, and HERC5 [4,5,6]. Apart from ubiquitinylation, human HERC5 and mouse HERC6 are also involved in a second PTM process, termed ISGylation, by utilizing the E3 ligases to catalyze the conjugation of ISG15, a ubiquitin-like protein induced by an antiviral cytokine interferon (IFN) [7,8]. Given the regulatory roles of HERCs-mediated ubiquitinylation and ISGylation in various physiological activities [9,10], four small HERCs exhibit differential potentials to inhibit HIV-1 particle production [1]. Notedly, HERC5 and HERC6 are transcriptionally induced in viral-infected cells [7,8,11]. These results highlight that small HERCs are involved in innate IFN antiviral immunity although the details are largely unknown.

IFN response is believed to begin with a rapid recognition of viral products by host pattern recognition receptors (PRR) including retinoic acid-inducible gene I (RIG-I)-like receptors (RLRs), cGAS (cyclic GMP-AMP synthase), and TLRs (toll-like receptors) [12]. Such recognition initiates distinct signaling cascades through the recruitment of downstream adaptors MAVS (mitochondrial antiviral signaling protein), STING (stimulator of interferon response cGAMP interactor 1, also known as MITA), or TRIF (TIR domain-containing adaptor protein inducing interferon-β), but finally converges on activating the TBK1 (TANK-binding kinase 1)-IRF (IFN regulatory factor) 3/7 signaling axis to turn on cellular IFN expression. The produced IFNs in turn induce the expression of hundreds of ISGs (IFN-stimulated genes), thus constituting the first line of defense against viral replication [12]. Despite the necessity and relevance of cellular IFN response, unregulated IFN expression is pathogenic and often fatal in mammals [13]. Therefore, some ISGs are induced to fine-tune the IFN expression [14,15,16], such as human HERC5 which promotes, as a typical ISG, cellular IFN expression through sustaining IRF3 activation [17].

It is documented that small HERCs originate from a common ancestor by gene duplication and chromosomal rearrangement [1,2,18], as evidenced by the findings that HERC3 and HERC4 generally display a strict 1:1 orthologous relationship across vertebrates, and they both reside in 2 chromosomes bearing the highest homology to each other [1,19]. However, we have recently revealed that only mammalian species have the orthologs of human HERC5 and HERC6, and nonmammalian vertebrates harbor a novel HERC7 subfamily that does not exist in mammals [19]. A *herc7* gene from crucian carp *Carassius auratus* is expressed as an ISG, and in viral-infected cells, it selectively targets three RLR signaling factors to alleviate IFN response by two distinct strategies [19]. Interestingly, varied copies of *herc7* genes are present in different fish species, with some unique to fish species [19]. This means that functional characterization of a fish species-specific *herc7* gene is of great significance for the delineation of fish species-specific antiviral immunity.

In this study, we found four *herc7* homologous genes (named HERC7a–d) in zebrafish Chromosome 1, which were transcriptionally induced in zebrafish tissues following SVCV (spring viremia of carp virus) infection. Phylogenetical analyses supported that the HERC7 subfamily has undergone species-specific expansion during the radiation of teleosts. We focused on zebrafish *herc7c* which was identified as an IFN-stimulated gene (ISG). Overexpression of zebrafish HERC7c promoted viral replication likely through the downregulation of fish IFN response. Mechanistically, zebrafish HERC7c targeted MAVS, STING, and IRF7 for protein degradation through the proteasomal-dependent pathway. Unlike the recently-identified crucian carp HERC7 showing the potential to conjugate ubiquitin and ISG15 to itself, zebrafish HERC7c has only E3 ubiquitin ligase activity, indicating function diversification of fish HERC7 family members in IFN antiviral response.

## 2. Results

### 2.1. Zebrafish Has Four herc7 Genes That Are Induced by SVCV Infection

Using crucian carp HERC7 as query, blast searches of zebrafish genome (GRCz11) identified four *herc7* homologous genes, sequentially named HERC7a (XP_17211386.1), HERC7b (XP_005160175.1), HERC7c (XP_021330683.1), and HERC7d (XP_005160166.1), which reside near the *herc3* gene locus in Chromosome 1 but not the *herc4* gene locus in Chromosome 13 (Figure 1A). Considering that the recently identified crucian carp HERC7 is transcriptionally induced by viral infection [19], we determined the expression patterns of four zebrafish *herc7* genes in response to SVCV infection. Intraperitoneal injection of SVCV into zebrafish resulted in increased mRNA expressions of zebrafish *ifnφ1* and *ifnφ3*, as well as *mxc* genes, a typical IFN-stimulated gene [20] (Figure 1B). Consistently, four zebrafish *herc7* genes were also transcriptionally induced by SVCV infection (Figure 1C). These results implied that SVCV infection activated a robust IFN response in all zebrafish tissues and also the expression of four zebrafish *herc7* genes.

### 2.2. Zebrafish herc7c Is a Typical IFN-Stimulated Gene

We tried to clone the full-length cDNA sequences of four zebrafish *herc7* genes. It was easy to obtain a single PCR band using a pair of primers designed against the 5′ UTR and 3′UTR sequences of the annotated *herc7c* in zebrafish genome data (left panel in Figure 2A). The cloned cDNA sequence contains the largest ORF of 2790bp, encoding a 929-aa HERC7c protein (Figure 2B). The annotated *herc7c* gene in zebrafish genome (GRCz11) was identified by automated computational analysis. Interestingly, it was annotated to contain 25 exons and was predicated to generate two transcript variants, encoding a 1002-aa protein and a 994-aa protein, respectively (Figure 2B). Sequence comparison revealed that our cloned *herc7c* cDNA sequence (OQ360721) does not contain the predicated 17th and 18th exons annotated in zebrafish genome (GRCz11) (Figure 2B). We repeated PCR experiments using different tissues samples; however, the same PCR product rather than both annotated transcript variants was cloned. Further, we designed a second pair of primes against sequences of the 16th and 19th exons (Figure 2B) to verify whether the 2 annotated variants were expressed. RT-PCR assays still obtained a single product with a predicated size of 199 bp representing our cloned *herc7c* cDNA sequence (OQ360721), without a larger product of 388 bp predicated from the 2 annotated transcription variants (right panel in Figure 2A). Thus, we thought that the cloned cDNA (OQ360721) might be the real transcript of the zebrafish *herc7c* gene.

To characterize whether zebrafish HERC7c is induced by IFN stimuli, we further cloned an 888-bp-5′ flanking sequence relative to the transcription start site of zebrafish *herc7c gene* (Figure 2C). This sequence has two putative ISRE/IRF-E sites, and it was next used to construct an *herc7c* promoter-driven luciferase plasmid (HERC7cpro-luc) and several derived mutants by mutating two predicated ISRE motifs (Figure 2D). As expected, the 888-bp-promoter of *herc7c* (HERC7cpro-luc) was robustly activated in EPC (*Epithelioma papulosum cyprini*) cells when transfected with poly(I:C) (polyinosinic-polycytidylic acid) as intracellular poly(I:C), or each of RLR signaling molecules, including RIG-I, MDA5, MAVS, STING, TBK1, IRF3, and IRF7 (Figure 2E). Overexpression of IFN also effectively activated HERC7cpro-luc (Figure 2E), and this activation was seen by individual or collective overexpression of STAT1, STAT2, and IRF9 (Figure 2F), three pivotal molecules involved in the IFN-triggered JAK-STAT signaling pathway [21,22,23].

The RLR pathway-induced IFN response can be triggered in fish cells by poly(I:C) transfection and SVCV infection [24,25]. Subsequent assays showed that intra-poly(I:C)-triggered activation of HERC7cpro-luc was severely decreased by individual overexpression of dominant negative mutants of RLR signaling factors (TBK1-K38M, IRF3-DN, and IRF7-DN) as well as JAK-STAT signaling factors (STAT1-ΔC and IRF9-ΔC) (Figure 2G). Compared with the full-length HERC7c promoter sequence (WT, −888~−1) that could be activated by intra-poly(I:C), extra-poly(I:C), or SVCV, a truncated promoter sequence (−248~−1) containing 2 putative ISRE motifs (ISRE1 and ISRE2) retained the intact promoter activation, but a second truncated promoter sequence devoid of 2 ISRE motifs (−888~−249) failed to respond to 3 stimuli above (Figure 2H). Mutation of ISRE1 but not ISRE2 gave a weakened luciferase activity (Figure 2I), indicating that the proximal ISRE1 motif rather than the distant ISRE2 was responsible for the HERC7c promoter activation. These results together indicated that zebrafish *herc7c* is a typical ISG.

### 2.3. Zebrafish HERC7c Inhibits Fish IFN Response

Given that the recently identified crucian carp HERC7 is involved in downregulation of the fish IFN response [19], we next investigated whether zebrafish HERC7c played a similar role. Luciferase assays showed that, compared with the control cells transfected with pcDNA3.1, poly(I:C) transfection resulted in robust activation of crucian carp IFN promoter-driven luciferase plasmid (CaIFNpro-luc); however, this activation was severely impeded by overexpression of zebrafish HERC7c in a dose-dependent manner (Figure 3A). Consistently, overexpression of zebrafish HERC7c in EPC cells inhibited poly(I:C)-triggered mRNA expression of *ifn* and *viperin* (Figure 3B), which was seen in a time-dependent manner (Figure 3C). These results indicated that zebrafish HERC7c functions as an inhibitor of IFN response in fish.

### 2.4. Zebrafish HERC7c Promotes Virus Replication in Fish Cells

To determine the effect of zebrafish HERC7c on viral replication, EPC cells were transfected with zebrafish HERC7c or pcDNA3.1 as control, followed by infection of SVCV at different titers. SVCV incubation yielded broad CPEs in zebrafish HERC7c-overexpressing cells compared with control cells (Figure 4A). Consistently, a higher transcription level of three SVCV genes (L, N, and G) was detected in HERC7c-overexpressing cells than in control cells (Figure 4B). Under the same conditions, the increase in the expression of cellular *ifn* and *viperin* in the presence of HERC7c overexpression was less than in the case of its absence (Figure 4C). Subsequent assays showed that overexpression of zebrafish HERC7c inhibited SVCV-triggered mRNA expression of cellular *ifn* and *viperin* over infection time (Figure 4D) and, concomitantly, resulted in a transcription elevation of the three SVCV genes (Figure 4E). These results indicated that zebrafish HERC7c promotes virus replication in fish cells likely through downregulation of the fish IFN response.

### 2.5. Zebrafish HERC7c Downregulates Fish IFN Response by Targeting STING, MAVS, and IRF7

Luciferase assays showed that overexpression of STING stimulated the activation of zebrafish IFNφ1 promoter- or IFNφ3 promoter-driven luciferase plasmids (DrIFNφ1pro-luc, DrIFNφ3pro-luc); however, this activation was alleviated in the presence of zebrafish HERC7c (Figure 5A). Zebrafish HERC7c-mediated alleviation was seen when IFN promoter activation was stimulated by MAVS or IRF7 (Figure 5B), but not by TBK1 or IRF3 (Figure 5C). It was noteworthy that overexpression of zebrafish IRF3 just activated zebrafish IFNφ1 promoter but not zebrafish IFNφ3 promoter (Figure 5C), similar to our previous results [24]. These data indicated that zebrafish HERC7c targets STING, MAVS, and IRF7 to downregulate the IFN antiviral response.

### 2.6. Zebrafish HERC7c Facilitates STING, MAVS, and IRF7 Protein Degradation to Downregulate IFN Response

To further determine the mechanism of how zebrafish HERC7c downregulated the STING-mediated IFN response, the effect of zebrafish HERC7c on gene transcription of *ifn* and *sting* was initially investigated. As anticipated, zebrafish STING-induced transcription of cellular *ifn* and *viperin* genes was significantly attenuated by zebrafish HERC7c over infection time (Figure 6A). However, zebrafish HERC7c did not have influences on *sting* gene transcription because a nearly identical pattern was detected between the cells overexpressing STING alone and the cells overexpressing STING and HERC7c together; the detected pattern was a pair of primers that amplifies mRNAs only from the transfected STING plasmid (Figure 6B). Conversely, Western blots showed that simultaneous transfection of HERC7c and STING resulted in a decrease in STING proteins compared with the single transfection of STING (Figure 6C).

Similar assays showed that overexpression of zebrafish HERC7c time-dependently alleviated mRNA expression of cellular *ifn* and *viperin*, which were triggered by MAVS or IRF7 (Figure 6D). Under the same conditions, zebrafish HERC7c did not make a difference on mRNA expression of the transfected *mavs* or *irf7* (Figure 6E) but significantly attenuated their protein expressions (Figure 6F). Zebrafish HERC7c-mediated protein degradation of STING, MAVS, and IRF7 was blocked by the addition of MG132 (an inhibitor of the ubiquitin-proteasomal-dependent degradation pathway) but not of chloroquine (an inhibitor of the autophagy-lysosomal-dependent degradation pathway) (Figure 7A–C). These results indicated that zebrafish HERC7c facilitates STING, MAVS, and IRF7 protein degradation to downregulate the IFN response.

### 2.7. Zebrafish HERC7c Is an E3 Ligase for the Conjugation of Ubiquitin but Not ISG15

Similar to the recently identified crucian carp HERC7 [19], zebrafish HERC7c contains an N-terminal RLD domain and a C-terminal HERC domain (Figure 8A). It is believed that the HECT domain entitles HERC proteins with the E3 ubiquitin ligase activity, and particularly, human HERC5 and mouse HERC6 are also responsible for protein ISGylation [9]. To this end, we investigated whether zebrafish HERC7c has a potential to transfer ubiquitin or ISG15. Similar to the recently-identified crucian carp HERC7 that can be ubiquitinated by itself [19], transfection of HEK293T cells with His-ubiquitin and HERC7c-Flag followed by affinity purification of His-ubiquitin using Ni^2+^-NTA resin revealed an enhanced level of ubiquitinated HERC7c proteins compared with overexpression of HERC7c alone (Figure 8B). Similar assays showed that crucian carp HERC7 rather than zebrafish HERC7c could be modified by ISG15 in cells when simultaneously transfected with ISG15 instead of ubiquitin (Figure 8C). These results indicated that zebrafish HERC7c is an E3 ligase responsible for ubiquitinylation but not for ISGylation.

### 2.8. Zebrafish HERC7c Is a Species-Specific Gene

The HERC7 subfamily exists in non-mammalian vertebrates [19]. To determine the relationship of zebrafish HERC7c in the HERC7 subfamily, we collected a total of 40 small HERC homologs from elephant shark (*C. milii*) to birds. Apart from four HERC7 members in zebrafish, blast searches of genome data revealed three HERC7 members in grass carp, three in common carp, and two in goldfish. Phylogenetical tree analyses showed that all HERC7 members are clustered into a clad that is distinct from the HERC3, HERC4, HERC5, and HERC6 subfamilies. Four zebrafish HERC7s were divided into two branches and each has no “one to one” orthologs in other fish species (Figure 9). These results indicated the occurrence of the species-specific expansion of the HERC7 subfamily in zebrafish, thus implying that zebrafish HERC7c might be a species-specific gene.

## 3. Discussion

HERC1 is the founding member of the HERC family characterized in human breast cancer cells [26], followed by the identification of a total of six HERC members in human, which are phylogenetically divided into large HERCs (HERC1 and HERC2) and small HERCs (HERC3-6) [2,10]. By genome-wide search of small HERC homologs from elephant shark to mammals and subsequently comprehensive evolutionary analyses, we provide evidence showing that, whereas HERC3 and HERC4 are conserved across vertebrates, the orthologs of human HERC5 and HERC6 are only present in mammals with a definite orthologous relationship to each other, and importantly, non-mammalian vertebrates have a unique HERC7 subfamily that might have been lost in modern mammals [19].

In this study, we cloned zebrafish *herc7c* cDNA by nested PCR. Interestingly, the obtained sequence is different from the two “annotated” transcripts in the zebrafish genome (GRCz11) by automated computational analysis. To verify whether our cloned sequence is a novel splicing variant of zebrafish *herc7c*, we designed a second pair of primers based on our hypothesis that, if there is a splicing variation, two PCR products will be amplified. However, only one band corresponding to the cloned sequence was obtained. Therefore, we think that our cloned cDNA represents the real transcript of the zebrafish *herc7c* gene.

Subsequently, we identified four *herc7* homologs in zebrafish Chromosome 1, showing that they are adjacent to *herc3*. In mammals, small HERCs generally reside in two gene loci, one containing a single *herc4* gene in one chromosome, and the other containing *herc3* and other *herc* genes in a second chromosome [1,19]. The chromosomal location link of zebrafish *herc3* and four *herc7* genes indicate that they might originate from a common ancestry. In addition, different fish species have varied copies of *herc7* genes, most of which seem not to exhibit a “one to one” orthologous relationship to each other. Given that teleost fish have additional whole-genome duplication [27,28,29], we propose that the varied *herc7* copies in different fish species might arise from segmental duplication after the additional whole-genome duplication. These results support a notion that the small HERC family has experienced gene duplication, chromosomal rearrangement, and gene loss events during vertebrate evolution [1,2,3,18].

The results in the current study establish that zebrafish HERC7 destabilizes different signaling molecules at the protein level to downregulate the IFN response during viral infection. First, zebrafish HERC7c is transcriptionally expressed along the viral infection. Secondly, zebrafish HERC7c benefits SVCV replication and concomitantly alleviates the IFN response. Thirdly, zebrafish HERC7c markedly abrogates the IFN promoter activation and *ifn* gene transcription by RLR signaling molecules, including STING, MAVS, and IRF7. Mechanistically, zebrafish HERC7c targets STING, MAVS, and IRF7 for proteasome-dependent protein degradation, as evidenced by the findings that this degradation is blocked by MG132. An interesting question is that zebrafish HERC7c targets IRF7 but not IRF3 to attenuate IFN expression, although IRF3 is most homologous to IRF7 [24,30,31]. This means a possibility that zebrafish HERC7c can specifically select the substrates to exert inhibitory effects. The same is true for the differential function of zebrafish IRF3 and IRF7 because they are responsible for the transcription of the zebrafish *ifnφ1* and *ifnφ3* genes, respectively [23,24]. Although vertebrate IRF3s exhibit a “one to one” orthologous relationship to each other, fish IRF3 is a virus- and IFN-induced protein, and instead, mammalian IRF3 is constitutively expressed even under viral infection [31,32,33]. These results further suggest that the inhibitory effect of zebrafish HERC7c on the IFN response might be a consequence of selection pressures that are exerted by fish viruses.

Given that zebrafish HERC7c is identified as an inhibitor of the fish IFN response, mammalian HERC5s play a positive role in limiting different virus replications [1]. In this paper, the authors present evidence that a coelacanth protein (XP_014354291.1) also displays a similar inhibitory effect on simian immunodeficiency virus replication [1]. However, subsequently we found that this coelacanth protein (XP_014354291.1) is indeed a HERC7 homolog although erroneously characterized as a coelacanth homolog of human HECR5 by comprehensive phylogenetic analyses [19]. Therefore, coelacanth does not contain an ortholog of human HERC5, and the coelacanth HERC7 protein (XP_014354291.1) acts as an intracellular viral inhibitor, similar to human HERC5 [1]. Moreover, the recently-identified crucian carp HERC7 attenuates the IFN response by utilizing two different mechanisms to degrade STING, MAVS, and IRF7 at the protein and mRNA levels [19]. These results indicate that the HERC7 subfamily members have undergone functional diversification, and thus they might play opposing roles in response to viral infection, such as the coelacanth protein (XP_014354291.1) as a positive regulator, and zebrafish HERC7c and crucian carp HERC7 as a negative regulator.

Small HERCs possess E3 ligase activity as a result of their C-terminal HECT domains [34]. Interestingly, whereas crucian carp HERC7 participates in both ubiquitinylation and ISGylation, zebrafish HERC7c just bears the potential to transfer ubiquitin (ubiquitinylation), further indicating that fish HERC7 subfamily members have experienced function diversification. In the present study, we did not investigate whether the E3 ubiquitin ligase activity of zebrafish HERC7c is tightly related to its inhibitory role in fish antiviral response; however, our previous report has shown that the inherent E3 ligase activity seems to not be required for the crucian carp HERC7 downregulation of the IFN response because a mutant without E3 ligase activity displays a nearly intact ability to alleviate the fish IFN response [19].

Given that four zebrafish *herc7* genes are generated by fish species-specific expansion of the HERC7 family, zebrafish *herc7c* might represent a zebrafish-specific gene, as evidenced by the fact that we failed to find a “one-to-one” ortholog of zebrafish HERC7c by comprehensive analyses of the available fish genome databases. Since zebrafish HERC7c expression is elevated along with viral infection, the data in the present study suggest that zebrafish HERC7c might mediate a fish species-specific regulation of the IFN response to avoid unregulated IFN production during viral infection. Further studies might focus on in vivo function clarification of zebrafish HERC7c. If the function blockade of zebrafish HERC7c were able to effectively improve zebrafish survival against viral infection, it would highlight a relevance of HERC7 subfamily genes, such as zebrafish HERC7c with an inhibitory role, in fish antiviral precision breeding by gene-editing technology [35].

## 4. Materials and Methods

### 4.1. Cells, Virus, and Zebrafish

*Epithelioma papulosum cyprini* cells (EPC) were cultured in medium 199 supplemented with 10% fetal bovine serum (FBS) at 28 °C in a humidified incubator containing 5% CO_2_. Human embryonic kidney 293T cells (HEK293T, ATCC (CRL-3216)) were cultured in DMEM supplemented with 10% FBS at 37 °C.

Spring viremia of carp virus (SVCV) was propagated in EPC cells and tittered, according to the method of Reed and Muench, by a tissue culture ID50 assay. EPC cells were infected with SVCV at a final concentration of 5 × 10^3^ TCID_50_/mL.

Zebrafish adults (2-month-old), with similar sizes and weights (body length: 3 cm; weight: 0.4 g), were raised in a single batch according to standard protocol [36], which was approved by the Animal Care and Use Committee of Institute of Hydrobiology, Chinese Academy of Sciences. For viral infection, zebrafish were intraperitoneally injected with SVCV at 1 × 10^8^ TCID_50_/mL (25 μL/fish). After injection, they were kept at 28 °C water for 48 h without feeding. The control group received the same dose of 0.9% normal saline. After 48 h, the zebrafish were euthanized by immersing in a mixture of ice water for 20 min, followed by sampling of tissues.

### 4.2. Gene Cloning, Database Mining, and Sequence Analysis

Four immune tissues (spleen, liver, head kidney, and body kidney), from a zebrafish adult infected for 48 h with SVCV, were mixed to extract total RNA for cDNA synthesis by TRUEscript RT MasterMix (PC5801, Aidlab, Beijing, China). The mixed cDNA was used as a template to clone the ORF of zebrafish *herc7c* by nested PCR method. Zebrafish genome DNAs were extracted by the Wizard Genomic DNA Purification Kit (Promega) for PCR amplification of the *herc7c* promoter. The primers were designed against the computational annotated *herc7c* sequence in the zebrafish genome data (GRCz11) (Table 1).

Using zebrafish HERC7c (GenBank accession no. OQ360721) protein sequence as a query, protein BLAST searches were performed on the genome databases of grass carp (*Ctenopharyngodon idella*), common carp (*Cyprinus carpio*), and goldfish (*Carassius auratus*). Three homologs of zebrafish HERC7 were found in grass carp (XP_051736083.1, XP_051739640.1, and XP_051731642.1) and common carp genomes (XP_042627096.1, XP_042617724.1, and XP_042617185.1), and two in goldfish genome (XP_026059385.1, XP_026059288.1). Meanwhile, 40 sequences of small HERC family members in other species, which were verified by evolutionary analysis in our previous study [19], were collected for subsequent evolutionary tree analysis. They include: HERC3 (pufferfish, elephant shark, and grass carp), HERC4 (mouse and zebrafish), HERC5 (hedgehog and bat), HERC6 (human and pika), HERC5/6 (striped catfish, coelacanth, elephant shark, frog, and green anole), and HERC7 (Atlantic salmon, river trout striped catfish, milkfish, zebrafish, common carp, grass carp, crucian carp, goldfish, elephant shark, coelacanth, green anole, and chicken). Multiple alignments were carried out with ClustalW2 to make a phylogenetic tree by neighbor-joining methods in Geneious. Transcription factor-binding sites were predicated using JASPAR database (http://jaspar.genereg.net/) accessed on 5 June 2022.

### 4.3. Plasmid Construction

For overexpression, the ORF of DrHERC7c was cloned into EcoR I and BamH I sites of pcDNA3.1/myc-His (-) A (Invitrogen). At the same time, different tags (HA, Flag) were added to the C-terminus of DrHERC7c by reverse amplification primers. For promoter analysis, 3 size-different 5′ flanking sequences of DrHERC7c including HERC7cpro-luc (−888–−1), HERC7cpro-luc (−888–−249), and HERC7cpro-luc (−248–−1) were cloned into Nhe I site of the pGL3-basic plasmid. The same method was carried out for three mutations plasmids (HERC7cpro-mut1, HERC7cpro-mut2, and HERC7cpro-mut1+2). Zebrafish plasmids DrMAVS/DrMAVS-Flag, DrSTING/DrSTING-HA, TBK1/TBK1-HA, IRF3/IRF3-HA, and IRF7/IRF7-HA were described in our previous reports [37,38]. Other plasmids including TBK1-K38M, IRF3-DN, IRF7-DN, IFNφ1pro-luc, IFNφ3pro-luc, STAT1-ΔC, and IRF9-ΔC were described previously [21,22,24,30].

### 4.4. Transfection and Luciferase Activity Assays

Cell transfection assays were performed with polyethylenimine, Linear (PEI, MW 25,000) (Sigma-Aldrich, Shanghai, China) according to our previous reports [15,36]. Luciferase activity assays were performed by a Junior LB 9509 luminometer (Berthold, Pforzheim, Germany) using Dual-Luciferase Reporter Assay System (Promega), as described previously [15,16,33]. The results were the representative of at least three independent experiments, each performed in triplicate. Luciferase activities were normalized to the amounts of internal control Renilla luciferase activities.

### 4.5. RNA Extraction, cDNA Synthesis, and Real Time-PCR

Cellular total RNAs were extracted by EASYspin Plus Kit (Aidlab, Beijing, China), followed by DNase treatment to remove residual DNA. First-strand cDNA was synthesized using MonScriptTM RTIII Super Mix with dsDNase Kit (Monad, Suzhou, China) according to the manufacturer’s protocol [36]. Real-time PCR (RT-qPCR) was performed with Hieff qPCR SYBR Green Master Mix (Yeasen, Shanghai, China) on the CFX96 real-time system (Bio-Rad). PCR condition was set by referring to the operation manual of the Hieff qPCR SYBR Green Master Mix. The relative expression was normalized to the expression of β-actin and represented as the fold induction relative to the expression level in the control cells that was set to 1. The primers used in this study are listed in Table 1. The primer designing principle follows a single amplification band and an amplification length between 100–250 bp.

### 4.6. Ubiquitination Assays and ISGylation Assays

For ubiquitination assays, HEK293T cells seeded in 10 cm^2^ dishes were transfected with the indicated plasmids. After 30 h, the transfected cells were lysed, and the cell supernatant was incubated with Ni^2+^-NTA His. Bind Resin (Novagen) at 4 °C overnight, followed by Western blotting using tag-specific Abs [14]. For ISGylation assays, HEK293T cells were transfected with zebrafish HERC7c-HA together with or without 5 μg Flag-ISG15. After 30 h, the cells were collected for Western blotting with tag-specific antibodies.

### 4.7. Statistical Analysis

Student’s *t*-test is applied for statistical analysis of the data derived from luciferase assays and RT-PCR assays. All quantitative experiments were performed with at least three independent biological repeats. (* *p* < 0.05; ** *p* < 0.01; *** *p* < 0.001, ns: no significant.)

## Figures and Tables

**Figure 1 ijms-24-04592-f001:**
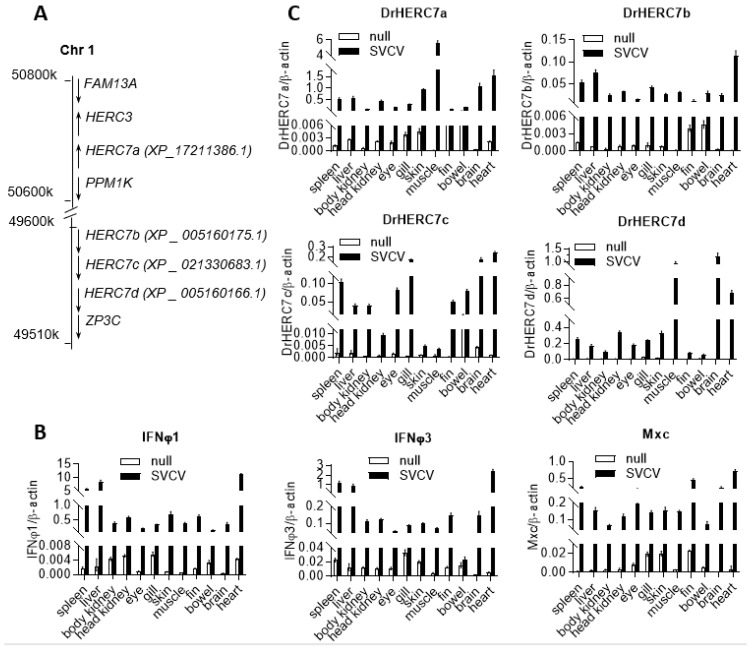
Zebrafish has four *herc7* genes that are induced by SVCV infection. (**A**) Schematic representation of zebrafish *herc3* gene locus containing four *herc7* homologous genes. (**B**,**C**) Four zebrafish *herc7* genes were transcriptionally induced in SVCV-infected zebrafish tissues. Zebrafish adults were intraperitoneally injected with SVCV for 48 h, followed by RT-PCR analyses of *ifnφ1*, *ifnφ3*, and *mxc* (**B**), as well as *herc7a-d* (**C**), in different zebrafish tissues.

**Figure 2 ijms-24-04592-f002:**
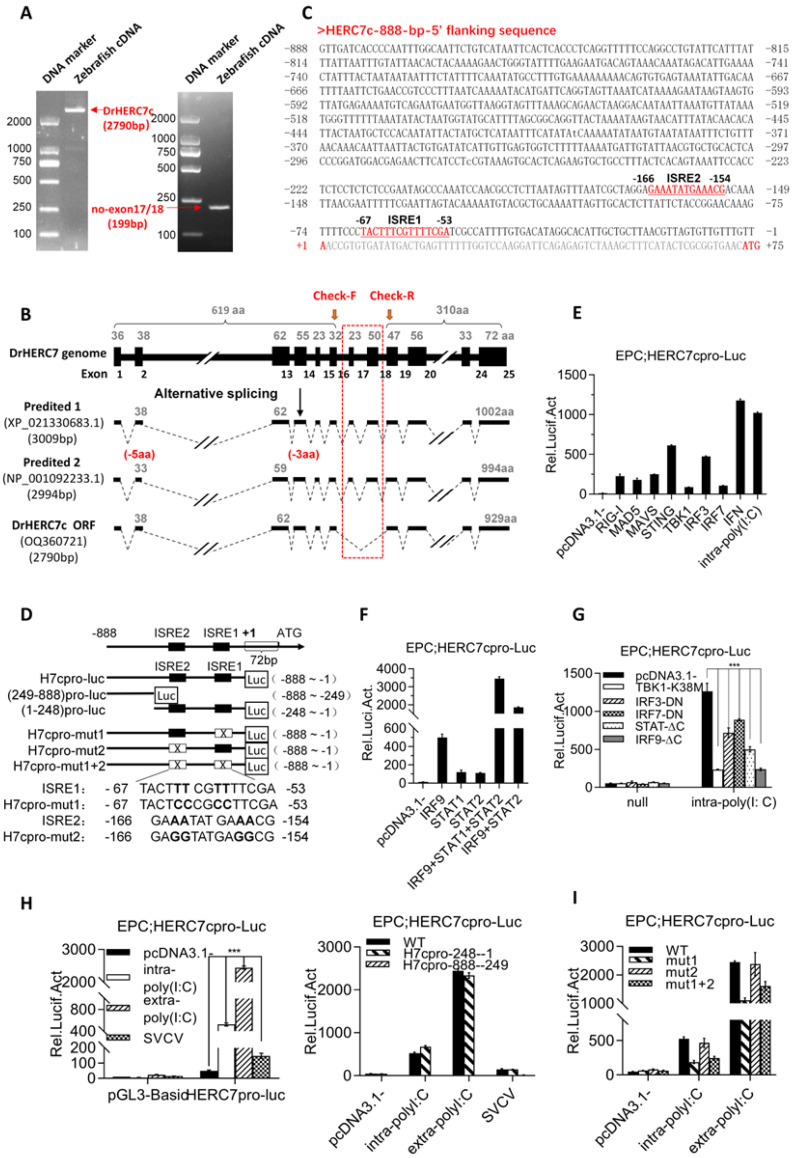
Zebrafish *herc7c* is a typical ISG. (**A**) Amplification of full-length cDNA (left) and fragment (right) of the *herc7c* gene. (**B**) Exon–intron structure and transcriptional splicing analysis of the zebrafish *herc7c* genome. The black box represents the exon. The red dotted box indicates the difference area among the possible spliceosomes. The red arrows mark the locations of the check primers. (**C**) The 5′ flanking sequence of zebrafish *herc7c*. Two ISRE motifs and the transcription start site are highlighted in red. Their respective positions are numerically marked in the sequence. (**D**) Schematic diagram of luciferase reporter plasmids under the control of 5′-flanking sequence of zebrafish *herc7c* gene and the derived truncates and point mutants. (**E**,**F**) Zebrafish HERC7c promoter was activated through the RLR signaling pathway and th eJAK-STAT pathway. EPC cells were co-transfected with 200 ng luciferase reporter plasmid (pGL3-Basic or HERC7cpro-Luc), together with 300 ng of each plasmid annotated in the figure. (**G**) Intracellular poly(I:C) stimulated-activation of zebrafish HERC7c promoter was blocked by overexpression of dominant negative mutants involved in the RLR pathway and the JAK-STAT pathway. (**H**) Zebrafish HERC7c promoter was activated by intra-poly(I:C), extra-poly(I:C), and SVCV. EPC cells were transfected with 200 ng HERC7cpro-Luc (left panel), or individually with HERC7cpro-Luc and 2 truncated promoter plasmids [(−888~−249) pro-luc, (−248~−1) pro-luc] (right panel). After 12 h, the cells were transfected again with 1 μg/mL poly(I:C) or directly incubated with 50 μg/mL poly(I:C) or infected with SVCV (final titer at 103 TCID50/mL). (**I**) ISRE motifs were responsible for zebrafish HERC7c promoter activation by intra-poly(I:C). EPC cells were transfected with HERC7cpro-Luc or with 3 ISRE-mutated luciferase plasmids (200 ng each). (*** *p* < 0.001).

**Figure 3 ijms-24-04592-f003:**
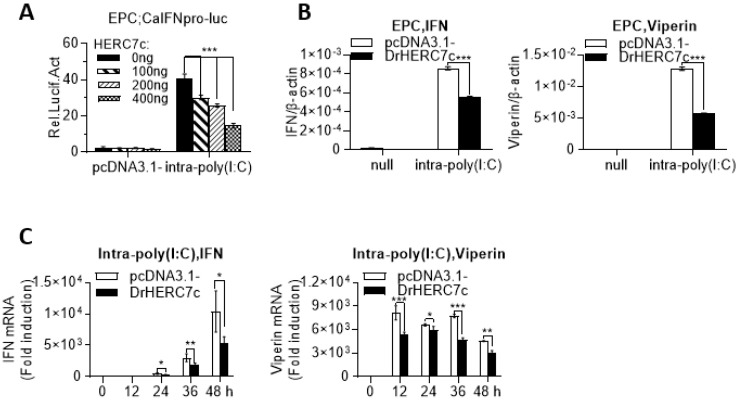
Zebrafish HERC7c suppresses intracellular poly(I:C)-triggered IFN response by luciferase assays. (**A**) Overexpression of zebrafish HERC7c inhibited fish IFN promoter activation by intracellular poly(I:C). EPC cells were cotransfected with 100 ng CaIFNpro-luc and increasing doses of zebrafish HERC7c for 24 h, followed by transfection with 1 μg /mL poly(I:C). (**B**,**C**) Zebrafish HERC7c inhibited mRNA expression of cellular *ifn* and *viperin* in EPC cells by intracellular poly(I:C). EPC cells were transfected with 2 μg of zebrafish HERC7c or pcDNA3.1- as control. After 12 h, the cells were transfected again with 1 μg/mL poly(I:C). At 36 h post infection (**B**) or at different time points (**C**), cells were sampled for RT-PCR analyses of cellular *ifn* and *mx* transcription. Error bars represent SD obtained by measuring each sample in triplicate. (*** *p* < 0.001, ** *p* < 0.01, * *p* < 0.05).

**Figure 4 ijms-24-04592-f004:**
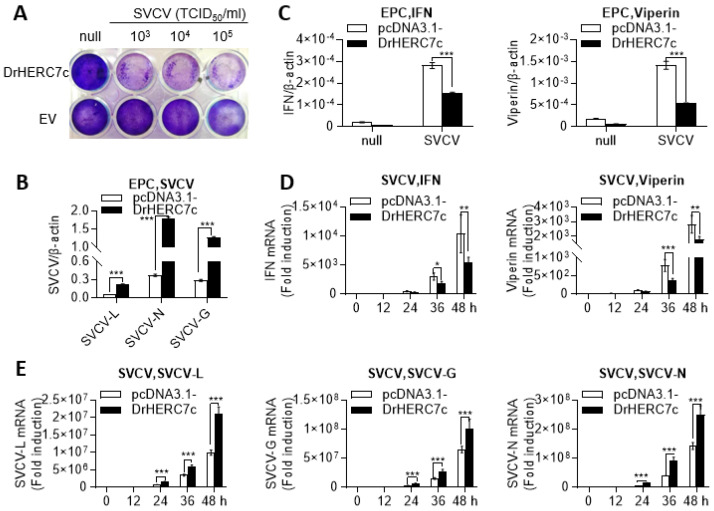
Zebrafish HERC7c promotes virus replication in fish cells. (**A**,**B**) Overexpression of zebrafish HERC7c promoted viral replication. EPC cells seeded in 24-well plates were transfected for 24 h with 500 ng HERC7c or pcDNA3.1 as control, followed by infection with SVCV at different titers (**A**) or at 1 × 10^3^ TCID_50_/mL (**B**). After 72 h, the cells were stained with crystal violet for detection of CPE (**A**), or 24 h later, the cells were collected for RT-PCR analyses of SVCV gene transcription. (**C**–**E**) SVCV replication was facilitated by overexpression of zebrafish HERC7c. EPC cells seeded in 24-well plates overnight were transfected for 12 h with HERC7c (0.5 µg) or pcDNA3.1(-) as control, followed by infection with SVCV. After 72 h (**C**), or at different time points (**D**,**E**), cells were sampled for RT-PCR detection of cellular *ifn* and *viperin* expression (**C**,**D**), and SVCV gene expression (**E**). (* *p* < 0.05, ** *p* < 0.01, *** *p* < 0.001).

**Figure 5 ijms-24-04592-f005:**
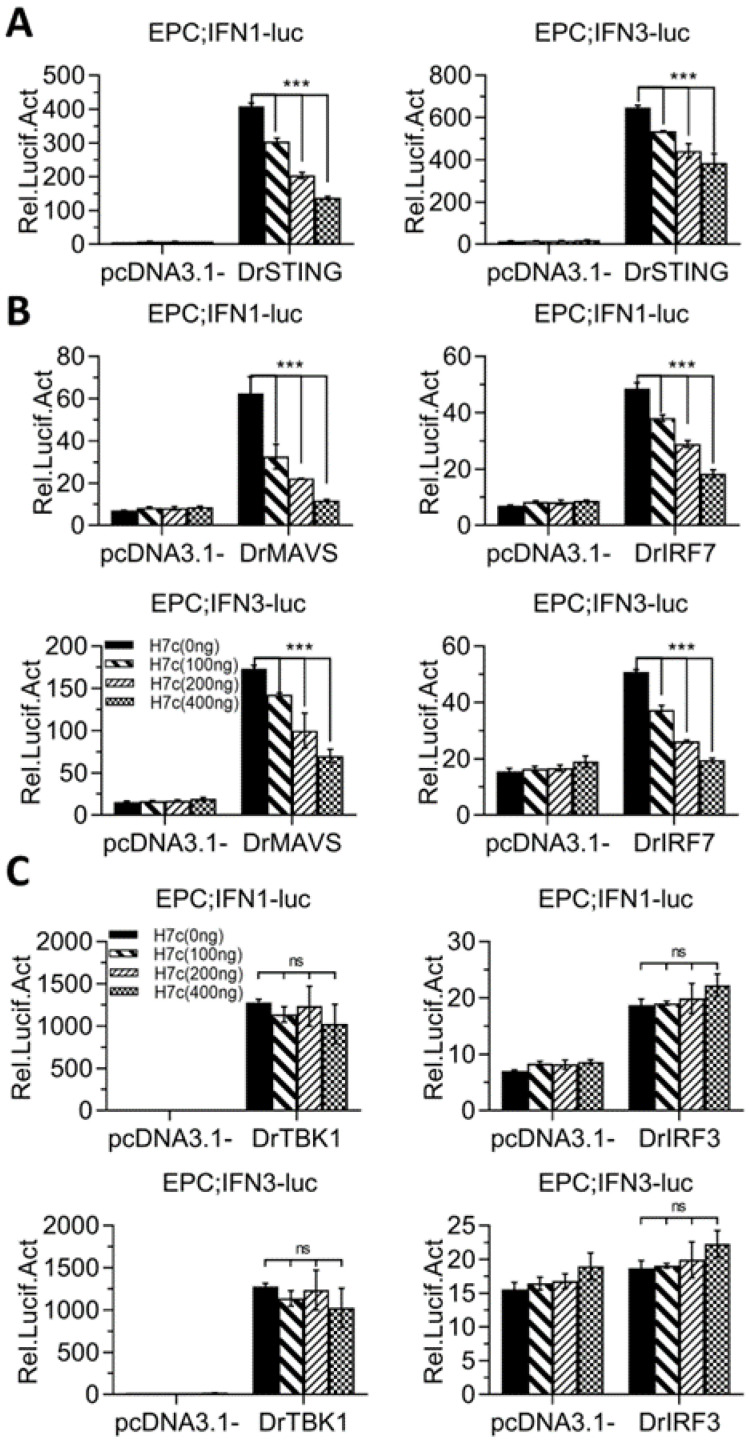
Zebrafish HERC7c suppresses the IFN response by targeting STING, MAVS, and IRF7. EPC cells seeded in 24-well plates were transfected with 100 ng DrIFNφ1pro-luc or DrIFNφ3pro-luc and increasing doses of HERC7c (0 ng, 50 ng, 100 ng, and 400 ng), together with 300 ng of zebrafish STING (**A**), MAVS or IRF7 (**B**), and TBK1 or IRF3 (**C**). After 24 h, the cells were collected for luciferase assays. Error bars represent SD obtained by measuring each sample in triplicate. (*** *p* < 0.001, ns: no significant).

**Figure 6 ijms-24-04592-f006:**
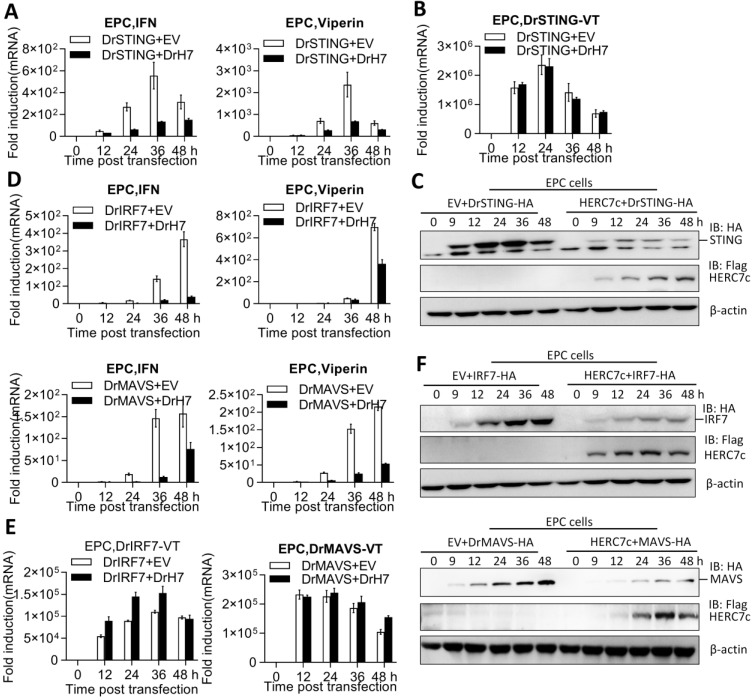
Zebrafish HERC7c attenuates STING, MAVS, and IRF7 protein levels to downregulate the IFN response.(**A**,**B**) Zebrafish HERC7c attenuated STING-directed *ifn* gene transcription but not *sting* gene transcription. EPC cells seeded in 3.5 cm^2^ dishes were cotransfected with zebrafish STING-HA and zebrafish HERC7c-Flag (1 μg each) for different time points, followed by RT-PCR analysis of cellular *ifn* and *viperin* (**A**) and the transfected *herc7c* and *sting* (**B**). Error bars represent SD obtained by measuring each sample in triplicate. (**C**) Zebrafish HERC7c attenuated zebrafish STING protein level. EPC cells seeded in 3.5 cm^2^ dishes were cotransfected with zebrafish HERC7c-Flag and zebrafish STING-HA (1 μg each) for different time points, followed by Western blotting analyses of STING protein. (**D**–**F**) Zebrafish HERC7c attenuated IRF7- or MAVS-directed *ifn* gene transcription. EPC cells were cotransfected as in (**A**,**B**) by replacing zebrafish STING with zebrafish IRF7 or MAVS, followed by RT-PCR analyses (**D**,**E**) or Western blotting (**F**).

**Figure 7 ijms-24-04592-f007:**
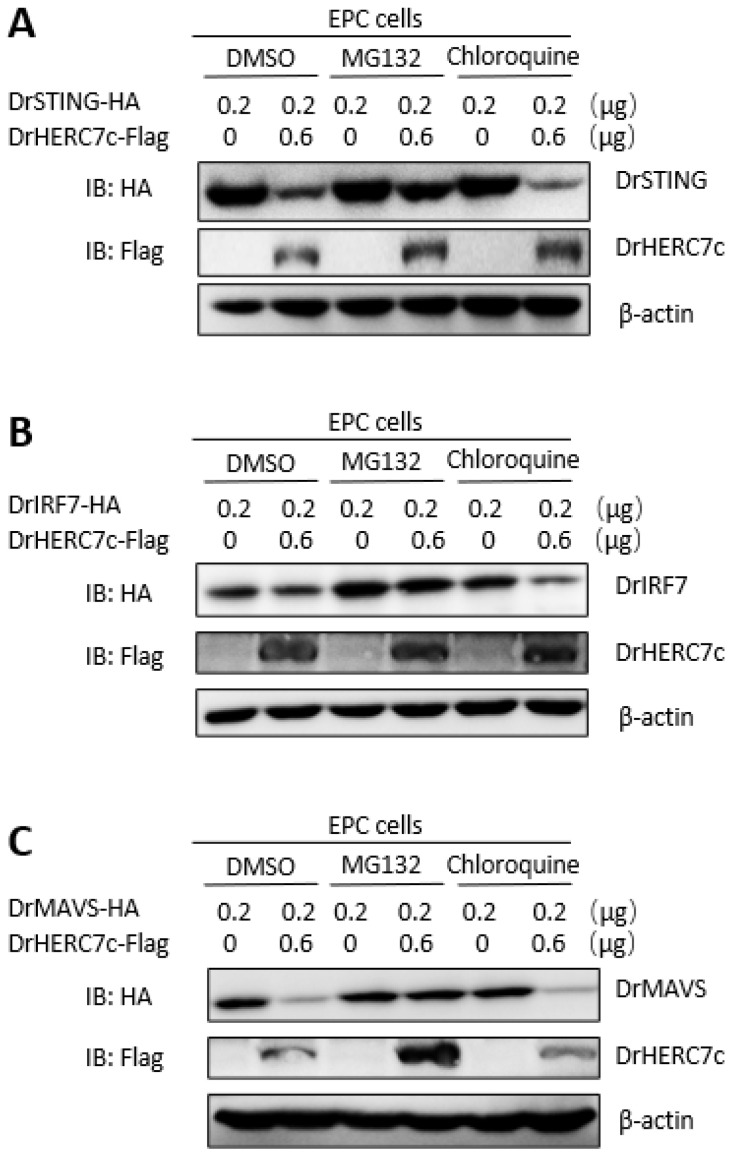
Zebrafish HERC7c-mediated degradation of STING, MAVS, and IRF7 was blocked by MG132 but not by chloroquine. EPC cells seeded in a 12-well plate were cotransfected with DrHERC7c-Flag and DrSTING-HA (**A**), DrMAVS-HA (**B**), or DrIRF7-HA (**C**), respectively. After 24 h, the cells were treated with MG132 (final concentration of 10 mM) and chloroquine (final concentration of 25 mM) for another 6 h, followed by Western blotting.

**Figure 8 ijms-24-04592-f008:**
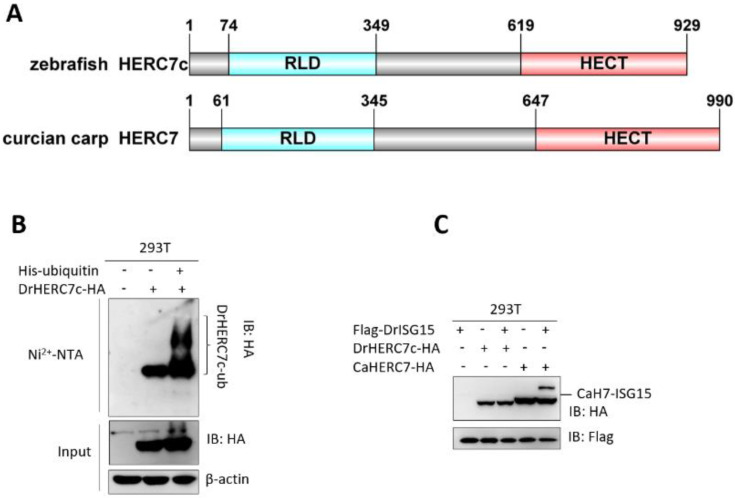
Zebrafish HERC7c is an E3 ligase for the conjugation of ubiquitin rather than ISG15. (**A**) Schematic diagram of zebrafish HERC7c protein and crucian carp HERC7 protein. (**B**) Polyubiquitination of zebrafish HERC7c. HEK293T cells plated in 10 cm^2^ dishes overnight were transfected with 5 μg zebrafish HERC7c-HA together with or without 4 μg His-Ub. After 30 h, cells lysates were incubated with Ni^2+^-NTA resin, followed by Western blotting. (**C**) ISGylation of crucian carp HERC7 but not of zebrafish HERC7c. HEK293T cells plated in 10 cm^2^ dishes were transfected with zebrafish HERC7c-HA or crucian carp HERC7-HA (5 μg each), together with or without 5 μg Flag-ISG15. After 30 h, the cells were collected for Western blotting.

**Figure 9 ijms-24-04592-f009:**
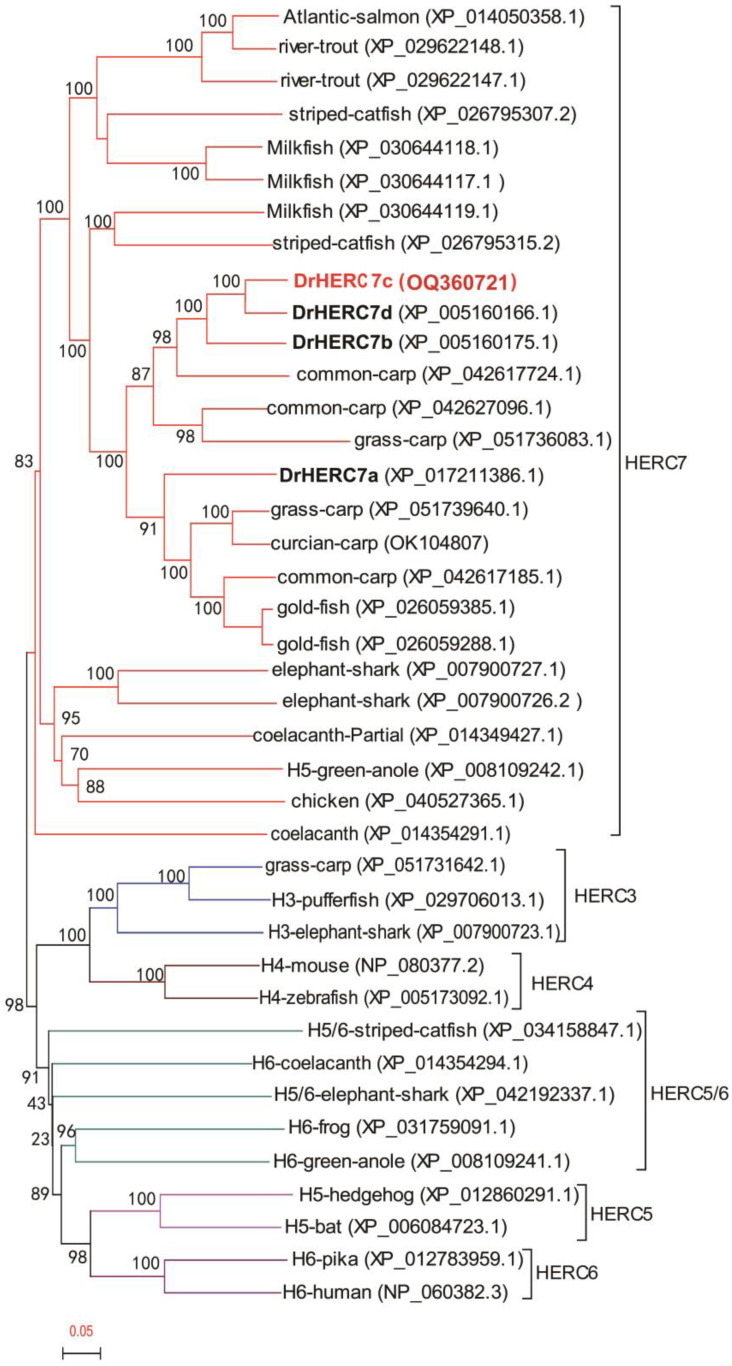
Phylogenetic evolutionary analysis of HERC7 subfamily containing zebrafish HERC7c. Phylogenetic trees were constructed using the neighbor-joining methods, and the bootstrap sampling was performed with 1000 replicates. The GenBank Accession Numbers of sequences are shown in brackets.

**Table 1 ijms-24-04592-t001:** The primers used in this study.

Primer Name	Primer Sequence	Usage
DrH7c-F1	CCGTGTGATATGACTGAG	Amplify herc7c ORF
DrH7c-F2	TAAAGCTTTCATACTCGCGG	
DrH7c-R1	GCAAAGTTTAGCAACAAACTTGAC	
DrH7c-R2	AGGCAAGTTCACCCTCGTCC	
Check-F	GTGCTTTGGATGGCATCGCGGC	Check primers
Check-R	CCAGTGAACTCTCATGAACTTCC	
DrH7c-pro-F	GTTGATCACCCCAATTTGG	Amplify 5′ flanking
DrH7c-pro-R	GCAAAGTTTAGCAACAAACTTGAC	
DrH7cpro-luc-F	GAGCTCTTACGCGTGGTTGATCACCCCAATTTGG	Construct luciferase plasmid
DrH7cpro-luc-R	TCGAGCCCGGGCTAGAACAAACAACACTAACGTTAAGC	
DrH7cpro-mut1-F	TACTCCCGCCTTCGA	
DrH7cpro-mut1-R	TCGAAGGCGGGAGTA	
DrH7cpro-mut2-F	GGAGAGGTATGAGGCGAC	
DrH7cpro-mut2-R	GTCGCCTCATACCTCTCC	
3.1-F	TGCTGGATATCTGCAGCCACCGCCACCATG	Construct eukaryotic expression plasmids
7c-FLAG-R	CGAGCTCGGATCGCTTATCGTCGTCATCCTTGTAATCTCGCCCTCGTCCAAAAACAGC	
7c-HA-R	CGAGCTCGGATCGAGCGTAGTCTGGGACGTCGTATGGGTATCGCCCTCGTCCAAAAACAG	
EPC-actin-Q-F	CAGATCATGTTTGAGACC	RT-PCR
EPC-actin-Q-R	ATTGCCAATGGTGATGAC	
EPC-Mx-Q-F	GGCTGGAGCAGGTGTTGGTATC	
EPC-Mx-Q-R	TCCACCAGGTCCGGCTTTGTTAA	
EPC-IFN-Q-F	ATGAAAACTCAAATGTGGACGTA	
EPC-IFN-Q-R	GATAGTTTCCACCCATTTCCTTAA	
EPC-Viperin-Q-F	AGCGAGGCTTACGACTTCTG	
EPC- Viperin-Q-R	GCACCAACTCTCCCAGAAAA	
VT-pcDNA3.1-Q-F	CGACTCACTATAGGGAGACC	
VT-DrSTING-Q-R	CCTTGAATGGAAGAGCAATTCCTC	
VT-DrMAVS-Q-R	CCCGATCAGAGATTGTGAGGCA	
VT-DrIRF7-Q-R	CTCGTTGATCTTGCCGCTGAC	

## Data Availability

All of the data referred to in this article are located within the main text. If there are requests for additional unpublished data not referred to in this article, please contact the corresponding author at ybzhang@ihb.ac.cn.

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
