# Peer review of "Zebrafish HERC7c Acts as an Inhibitor of Fish IFN Response"

_ijms, 2023, doi:10.3390/ijms24054592_

Round 1

Reviewer 1 Report

The authors investigate the role of Zebrafish HERC7c, an E3 ubiquitin ligase, in the regulation of fish IFN response. The main hypothesis the authors attempt to support is that Zebrafish HERC7c is a negative regulator of fish IFN antiviral response by targeting STING, MAVS and IRF7 for ubiquitination and degradation. Overall, the data presented are novel. The logic and hypotheses are presented clearly and flow well, the conclusions drawn from the data are reasonable. The paper is suitable for publication in IJMS after minor revisions.

Minor comments

1、Why were EPC cells used in this study? Why didn't the authors used cell line derived from zebrafish?

2、As shown in Figure 5, zebrafish HERC7c suppresses IFN response by targeting STING, MAVS and IRF7. What about the relationship between HERC7c and STING, MAVS and IRF7? Whether HERC7c interacts with STING, MAVS and IRF7 directly or indirectly? Which should be discussed in the discussion section.

 Author Response

Point 1: Why were EPC cells used in this study? Why didn't the authors used cell line derived from zebrafish?

 Response 1: Thank you very much. First of all, two zebrafish cell lines, ZF4 and ZFL, exihibit lower level of transfectiontion efficiency than EPC. Second, EPC cell lines, as cultured fish cells, have been preserved and used in many labs overy the world. A large number of previous studies have proved that it is feasible to use this cell line to study the function of fish genes.

Point 2: As shown in Figure 5, zebrafish HERC7c suppresses IFN response by targeting STING, MAVS and IRF7. What about the relationship between HERC7c and STING, MAVS and IRF7? Whether HERC7c interacts with STING, MAVS and IRF7 directly or indirectly? Which should be discussed in the discussion section.

Response 2: Thank you very much. Our results showed that zebrafish HERC7c targets STING, MAVS and IRF7 for protein degradation, thus impairing cellular IFN response. Since zebrafish HERC7c is an E3 ligase, we think that there is interaction of HERC7c with STING, MAVS and IRF7 directly, which is disscussed in the revised version as suggested.

Reviewer 3 Report

Overall, the research manuscript is well-prepared. However, make minor revision as per the comments provided in relevant sections.

 General comments:

·         All scientific name should be present in Italics.

·         Write about statistical methods performed in this study?

 Materials and methods:

4.1. Cells, virus and zebrafish

·         Were the zebrafish collected from single batch? Mention their size and weight.

·         Is it possible to inject 25 ml/fish with SVCV? I think it may be 25 μl/fish. After injection how the fish were maintained for 48 hrs?

·         How the fish were euthanized? Write the name of anesthesia agent with applied dose.

 4.6. RNA extraction, cDNA synthesis, and real time-PCR

·         Mention about primer designing protocol with their amplification size. Were they designed for this study or taken from previous research paper? 

·          Include the detail condition of real time PCR? 

·         How the results were expressed? Was the 2-ΔΔCT method used?

 Author Response

Round 2

Reviewer 2 Report

The authors did not address any of the major revisions that I judged necessary for the publication.  My decision remains the same as the previous revision and I cannot accept the manuscript in its present form for publication.